# The Sustainable Development Path of the Gold Exploration and Mining of the Sanshan Island-Jiaojia Belt in Laizhou Bay: A DID-SVAR Approach

**Sheng Zhang [1], Guoxiang Han [2], Ran Yu [1], Zuhui Wen [1], Meng Xu [3] and Yifu Yang [1,*]**

[1] School of Environment & Natural Resources, Renmin University of China, Beijing 100872, China; zhangs0531@ruc.edu.cn (S.Z.); 15605163260@163.com (R.Y.); zuhuiwen@163.com (Z.W.)
[2] State Key Laboratory of Urban and Regional Ecology, Research Centre for Eco-Environmental Sciences, Chinese Academy of Sciences, Beijing 100085, China; gxhan_st@rcees.ac.cn
[3] Department of Mathematics, School of Science, Beijing Jiaotong University, Beijing 100044, China; 18118021@bjtu.edu.cn
* Correspondence: yyang1991@ruc.edu.cn

**Abstract:** Gold is a vital strategic resource, and it plays an irreplaceable role in maintaining national financial security, enhancing currency guarantee capabilities, and serving as a country's last means of payment. Gold plays an essential role in several fields that are vital to sustainable development. In 2020, an ultra-large-scale gold deposit spanning land and sea was discovered in Sanshan Island-Jiaojia Belt, Laizhou Bay, China. Its owner, Shandong Gold Group, also established Sanshan Island as a new ecological mine model. Applying a difference in differences-structural vector autoregression (DID-SVAR) approach, our research found that the whole biodiversity of Laizhou Bay decreased by 0.27% purely due to gold exploration in Sanshan Island-Jiaojia. In the long run, gold mining will have an apparent 2.9% adverse effect on marine products, and fishing for marine products will have a 2.1% adverse effect on marine products themselves.

**Keywords:** gold mining; marine ecology; smelting pollution; sustainable development goals

## 1. Introduction

Gold reserves play a unique role in stabilizing the national economy, curbing inflation, and improving international creditworthiness. Gold has long been used as a safe-haven asset from the global economic perspective and has often been regarded as an alternative investment tool for US long-term Treasury bonds [1]. Gold has its special industrial needs. For example, 5G networks have high equipment performance and reliability requirements, and gold has increasingly become an indispensable conductor in the equipment manufacturing process [2]. This has provided a considerable boost to the increase in gold consumption in the wireless communication field and is expected to become the primary source of growth for the gold industry in the next few years. In addition, China is the world's largest gold jewelry market, gold producer, and gold consumer [3]. As China continues to urbanize, the 400 million people in the middle class will have more disposable income and limited investment opportunities, and their demand for gold will grow.

Gold mining is also inseparable from the achievement of the sustainable development goals (SDGs). For instance, the World Gold Council released a report entitled "Gold Mining's Contribution to the Sustainable Development Goals of the United Nations" [4]. Gold mining companies can make significant contributions to realizing the SDGs, especially in the four thematic areas of global partnership, namely social inclusion, economic development, responsible mining, and energy and environment, to bring about positive changes. For example, members are committed to being effective in multi-stakeholder partnerships, conscientiously improving local social inclusiveness in mining countries, supporting the socio-economic development of their mineral sources, and making significant progress

in energy and water conservation. Considering the wide range of countries involved in the gold mining industry, it is essential to make gold mining companies achieve these global goals.

On 24 December 2020, at a press conference held by the Information Office of the Shandong Provincial Government to introduce the situation of Shandong's mining breakthrough strategy, the Shandong Provincial Department of Natural Resources Jiaodong area became the world's third-largest gold mining area [5]. In particular, the discovery of three thousand-ton goldfields, including Sanshan Island, has made Jiaodong the third-largest gold mining area globally and has consolidated Shandong Province's position as the nation's most extensive gold production base. In the Jiaojia and Sanshan Island mining areas, several shallow gold deposits were previously thought to be distributed independently. In contrast, the central ore bodies extending to the deep gold ore bodies were connected or superimposed [6]. Experts pointed out that the Sanshan Island sea area gold mine was one of the few super-large gold mines globally, with the enormous scale and the most reserves in single domestic mines. In the Jiaodong area, affected by the early Cretaceous crust-mantle mixed-type large-scale deep magma uplift, the crust-derived exquisite rock base formed in the Jurassic uplifted strongly. The overlying metamorphic rock strata were primarily detached, creating demolition. This kind of hot uplift-extensional structure provided favorable conditions for the formation of gold deposits. Magmatic activities provided suitable sources, heat sources, and fluids for the construction of gold deposits, and extensional detachment structures offer suitable space for forming gold deposits [7]. The coupling of magma and fluid caused large-scale metallogenesis of Jiaodong-type gold deposits.

It is noteworthy that Laizhou Bay is located in the southern part of the Bohai Sea and the northern part of the Shandong Peninsula (Figure 1, charted by Arcgis 10.2). It is the largest semi-enclosed bay in the Bohai Sea. The water depth in the basin is relatively shallow, and the Yellow River brings many nutrients. It is one of the spawning and feeding grounds for the three major marine organisms in the Yellow and Bohai Seas. The coastline of Laizhou Bay and the sea area belong to Kenli County, Dongying City, Shandong Province, Shouguang City, Hanting District, Weifang Laizhou City, Zhaoyuan, and Yantai City.

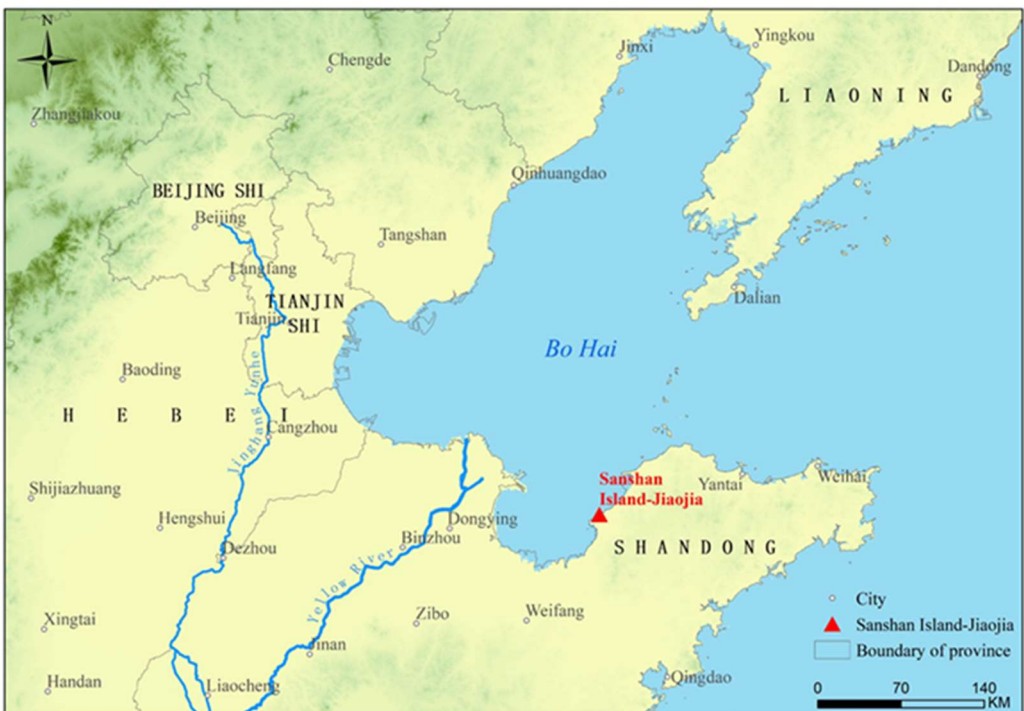

**Figure 1.** Laizhou Bay. The red triangle mark is where Sanshan Island-Jiaojia is located.

The Sanshan Island mining area is covered by seawater, and the central area has a water depth of 8.5–20 m [6]. The bedrock is mainly composed of Jurassic Linglong-type granite and Neoarchean meta-gabbro. The thickness of the Quaternary deposits under seawater is generally 35–40 m. They are the coarse, medium, and fine sand and silt of marine sediments.

According to Article 3(1) of the Chinese Mineral Resources Law, mineral resources belong to the state, and the State Council exercises state ownership of mineral resources. The state ownership of mineral resources on the surface or underground shall not change due to the land ownership or use rights on which it depends. Since the state monopolizes China's gold production, obtaining a specific production capacity is not easy. Fortunately, Shandong Gold Mining Co., Ltd., a subsidiary of the state-owned Shandong Gold Group, was listed on the Shanghai Stock Exchange in 2003, making it the second listed company in the national gold industry. As an essential asset, Sanshandao-Jiaojia Gold Mine should be disclosed following the corresponding disclosure rules, which allows us to obtain or estimate its output.

This manuscript is one of the very few pieces of literature, if not the first, quantitatively describing the pollution of marineenvironment by marine mineral mining. This will contribute to the 2022 United Nations (UN) Ocean Conference held in Lisbon, Portugal, and the accomplishment of Sustainable Development Goal (SDG) 14. Based on the DID-SVAR approach, we found that the impact of pollution caused by gold exploration activities and mining activities on marine ecology still cannot be ignored. Gold mining companies should control the significant environmental, social, and governance (ESG) risks associated with gold mining to achieve the SDGs.

## 2. Materials and Methods

### 2.1. Data Processing

We collected data from the former China Marine Statistical Yearbook and China Marine Ecological Environment Status Bulletin (2011–2020), Shandong Gold Group's listing announcements (2011–2019), Yantai Statistical Yearbook (2011–2019), and Laizhou Statistical Bulletin (2011–2019, Laizhou is a county-level city of Yantai City), etc. The four indicators we selected were "marine biodiversity index" (from the former China Marine Statistical Yearbook and China Marine Ecological Environment Status Bulletin), "gold production" (from Shandong Gold Group's former Laizhou Subsidiary and Sanshan Island Gold Mine), "precious metal smelting output values" (primarily gold, from Laizhou Statistical Bulletin), and "marine product production" (2011–2019) (from Yantai Statistical Yearbook).

The marine biodiversity index came directly from the former China Marine Statistical Yearbook and the China Marine Ecological Environment Status Bulletin, average weighted by three indicators (indexes consist of "phytoplankton", "zooplankton", and "macrobenthos", averaged by the weights 0.33, 0.33, 0.33). The missing values were filled with the geometric mean of the previous and subsequent years. If it was in the first or last years, the mean value of all the years was used. It is noteworthy that the original data provider, the State Oceanic Administration, was merged into the Ministry of Natural Resources after 2018. From 2019, the new biodiversity data provider, the Ministry of Ecology and Environment, would only calculate the marine biodiversity index in the summer instead of the whole year.

Gold production data were obtained from the listing announcement of Shandong Gold Group, excluding subsidiaries' yields in unrelated regions according to accounting standards. In addition, Shandong Gold Group did not disclose Sanshan Island-Jiaojia's production until 2017. However, as a monopoly, it clearly stated in almost all of its listing announcements that it would stabilize its output and control the proportion of different mining areas. Considering that the proportion of Sanshan Island-Jiaojia's gold production in Shandong's total gold production became stable at around 53–54% after 2017, we used 53% to estimate Sanshan Island-Jiaojia gold production from the Shandong Gold Group's total gold output before 2017.

The precious metal smelting output values and output of marine products were obtained from the Laizhou Statistical Bulletin and Yantai Statistical Yearbook because in the marine environment, Laizhou's sewage cannot only affect Laizhou's marine ecological environment. To a certain degree, we used the precious metal smelting output values of Laizhou as an instrumental variable for pollution compared to gold production.

Except for the marine biodiversity data index, all data are first-order logarithmic difference processed to maintain data stationarity and removal dimension. In addition, in the case of a small growth rate, the first-order logarithmic difference mathematically approximates the growth rate. We used Stata 17 to proceed with the further analysis.

### 2.2. Main Pollutants

According to the information disclosed by the local environmental supervision, the annual reports after the audit of the listed company, and the corporate social responsibility reports, the main types of pollution were COD and nitrogen oxides [8]. The Sanshan Island-Jiaojia gold mine belt is located in the central area of Laizhou Bay. Its wastewater is mainly discharged to Laizhou Bay and directly affects Laizhou, Yantai, and other cities. Shandong Gold Mining Group handles heavy metals produced in traditional gold production processes such as sulfur, arsenic, mercury, cyanide, copper, lead, and zinc. Still, the local standards for chemical oxygen demand are relatively lenient. The chemical oxygen demand is often used as a standard to measure organic matter in the water. The greater the chemical oxygen demand, the more seriously the water body is polluted by organic matter and the less oxygen there is. It should be noted that the average annual COD emission concentration reached 22.79 mg/L according to the 2020 listed company annual report.

In addition, the process of gold production, dissolution, and reduction require chemical oxidants and agents such as nitric acid, hydrochloric acid, and sulfite. It also involves using oxidants (pure oxygen or oxides) to accelerate the leaching process, all of which will produce COD. In gold production, aqua regia, nitrogen oxides, and acidic exhaust gases (NOX, HCl, and HNO$_3$) are generated in the Regia dissolution process, which enter the water body resulting in eutrophication [9–12]. All of these will lead to water hypoxia and PH changes, which will affect benthic organisms, plankton, fish fry, etc., and ultimately lead to the death of marine life.

### 2.3. DID Method

In an ideal experimental design, the decision of the experimental group and the control group members is entirely random. Therefore, the allocation of individuals can be considered random, utterly independent of individual characteristics or other factors that may affect the experiment results. Thus, the explanatory variable is not related to the missing disturbance items. In this way, omitted variable bias or endogeneity bias are avoided. Natural experiments or quasi-experiments have similar specialties. Due to some external emergencies that did not occur for experimental purposes, the parties seemed to be randomly divided into the experimental group or the control group. For example, a local area passed a policy, but another neighboring area did not. People in the two provinces did not know in advance which provincial capital adopted this policy. In this way, from investigating the effect of this policy, it can be approximately considered that the people are randomly assigned to live in a certain province, or randomly assigned to a certain group, namely the experimental group (passed the policy) and the control group (no policy). This situation is called a "natural experiment" or "quasi-experiment" [13–15].

In order to change the explanatory variable before and after the experiment, consider the following two periods of panel data:

$$y_{it} = \alpha + \gamma D_t + \beta x_{it} + u_i + \varepsilon_{it} \ (i = 1, \cdots, n; t = 1, 2) \tag{1}$$

$D_t$ is the experimental period dummy variable ($D_t = 1$, if $t = 2$, after the experiment; $D_t = 0$, if $t = 1$, before the experiment ), and $u_i$ is the unobservable individual variable and the policy dummy variable (policy dummy ) is:

$$x_{it} = \begin{cases} 1, & \text{if } i \in \text{ experimental group, } t = 2 \\ 0, & \text{other} \end{cases} \tag{2}$$

Therefore, when $t = 1$ (the first period), the experimental and control groups are not treated differently; $x_{it}$ is equal to zero. When $t = 2$ (the second period), the experimental group $x_{it} = 1$, while the control group is still equal to zero. Since we use two-period panel data, Equation (1) can be a first-order difference to eliminate $u_i$,

$$\Delta y_i = \gamma + \beta x_{i2} + \Delta \varepsilon_i \tag{3}$$

Using OLS to estimate the above formula, a consistent estimate can be obtained. According to the same reasoning as the difference estimator (differences estimator), therefore, this estimation method is called the "difference-in-differences estimator" (DID), the average change in the experimental group and the average change in the control group. The difference is recorded as $\widehat{\beta}_{\text{DID}}$. Therefore, the DID estimator has eliminated the influence of the "pre-experiment difference" between the experimental group and the control group as much as possible [15].

### 2.4. SVAR Method

Sims proposed the VAR model to analyze the dynamic relationship between economic variables as a solution [13,16]. However, the impulse response function of the simplified VAR depends on the order of the variables, and the simplified VAR cannot reveal the economic structure. For this reason, economists tried to re-incorporate the structure into the VAR model, allowing current influences between variables to form a "structural VAR" method [13,17–20].

Consider the general form of SVAR. Starting from the simplified VAR of order $p$:

$$y_t = \Gamma_1 y_{t-1} + \cdots + \Gamma_p y_{t-p} + u_t \tag{4}$$

$y_t$ is an M $\times$ 1 vector; $u_t$ is a simplified disturbance term, allowing contemporaneous correlation. Multiply a non-degenerate matrix $A$ on both sides of Equation (4) at the same time:

$$A y_t = A \Gamma_1 y_{t-1} + \cdots + A \Gamma_p y_{t-p} + A u_t \tag{5}$$

After shifting the items, we can obtain:

$$A \left( I - \Gamma_1 L - \cdots - \Gamma_p L^p \right) y_t = A u_t \tag{6}$$

We hope that the disturbance terms of SVAR are orthogonal. A simple way is to set $A u_t = \varepsilon_t$, where $\varepsilon_t$ is the structural disturbance term of SVAR, and there is no contemporaneous correlation; but this assumption may be too strong (matrix $A$ is derived from economic theory on economic structure modeling; at the end, it must be able to simultaneously make $A u_t$ irrelevant). More generally, suppose $A u_t = B \varepsilon_t$, where B is an M $\times$ M? matrix; then Equation (6) can be written as

$$A \left( I - \Gamma_1 L - \cdots - \Gamma_p L^p \right) y_t = A u_t = B \varepsilon_t \tag{7}$$

The covariance matrix of the structural disturbance term $\varepsilon_t$ is standardized to the identity matrix $I_M$. Equation (7) is called the "AB model" (AB-Model) of SVAR (Amisano and Giannini, 1997). For the AB model, the focus of the analysis is on the effects of orthogonalization shocks, so the structural disturbance terms are generally assumed $\varepsilon_t$

orthogonal. Multiply both sides of Equation (7) by $A^{-1}$, and the corresponding simplified VAR can be obtained:

$$y_t = \Gamma_1 y_{t-1} + \cdots + \Gamma_p y_{t-p} + \underbrace{A^{-1}B\varepsilon}_{u_t} \tag{8}$$

Since $u_t = A^{-1}B\varepsilon_t$, the covariance matrix of the simplified perturbation term $u_t$ is:

$$\text{Var}(u_t) = A^{-1}BB'A^{-1'} \tag{9}$$

For the structural VAR model (9), the total number of parameters to be estimated is $M^2$ (the number of parameters of matrix $A$)$+M^2$ (the number of parameters of matrix $B$) $+pM^2$ (the number of parameters of matrix $\Gamma_1, \cdots, \Gamma_p$ the number of parameters), that is, $2M^2 + pM^2$.

Therefore, to recognize the AB model (7), at least $\left[2M^2 - M(M+1)/2\right]$ constraints must be imposed on the elements in the matrices $A$ and $B$. Even if the main diagonal elements of matrix $A$ are standardized to 1, it still needs to add $\left[2M^2 - M - M(M+1)/2\right]$ additional constraints. If so many constraints are applied, it is just recognition; if more constraints are applied, it is over-recognition. This order condition (order condition) is necessary for identifying the AB model.

In order to estimate the SVAR model, it is generally assumed that the structural disturbance term $\varepsilon_t$ obeys a multi-dimensional normal distribution, that is, $\varepsilon_t \sim N(0, I_M)$, and then the maximum likelihood estimation with constraints. Although this MLE estimator is derived under the assumption of multi-dimensional normality, the QMLE estimator is still consistent under weaker conditions.

## 3. Results

### 3.1. DID Analysis

We selected 11 national key monitoring bays regarding marine biodiversity, including Laizhou Bay taking 2015 as the dividing line, the initial large-scale exploration year (7), as the treatment group, and the remaining ten bays as the control group. Table 1 gives the estimated average treatment effect on treatment (ATET), indicating that the biodiversity of Laizhou Bay has decreased by 0.27 percentage points compared with the bay where gold exploration has not been implemented [21]. One basic premise of the DID is that if the processing group is not affected by external shocks, its trend should be the same as the control group (so the latter can be used to control the time effect). This is the so-called parallel trend. Figure 2 below visually shows the idea of DID and the test of the parallel trend hypothesis.

The marine biodiversity indicators calculated by the former State Oceanic Administration and the current Ministry of Ecology and Environment mainly involve benthic organisms and plankton. According to reports, around 2015, China's first offshore gold exploration project deployed 44 boreholes in 17 square kilometers of Sanshan Island, Laizhou, with a most profound hole depth of more than 2000 meters. According to reports, the entire sea area of Sanshan Island in Laizhou City has been drilled for more than 120,000 meters. During the peak construction period, 52 sets of rigs were launched, 67 offshore drilling platforms were built, and construction personnel reached more than 1000 at most [5,7]. Although partly recovered in 2016 and subsequent years, the marine biodiversity index has not yet returned to its pre-2015 level, except for the outliers caused by changes in the statistical caliber in 2019. The impact on the local marine biodiversity is not negligible.

**Table 1.** DID regression results.

| Total | Coefficient | Robust HC2 Std. Err. | t | P > t | (95% Conf. | Interval) |
|---|---|---|---|---|---|---|
| ATET (1 vs. 0) | −0.2697147 | 0.0769002 | −3.51 | 0.007 | −0.44367 | −0.09575 |

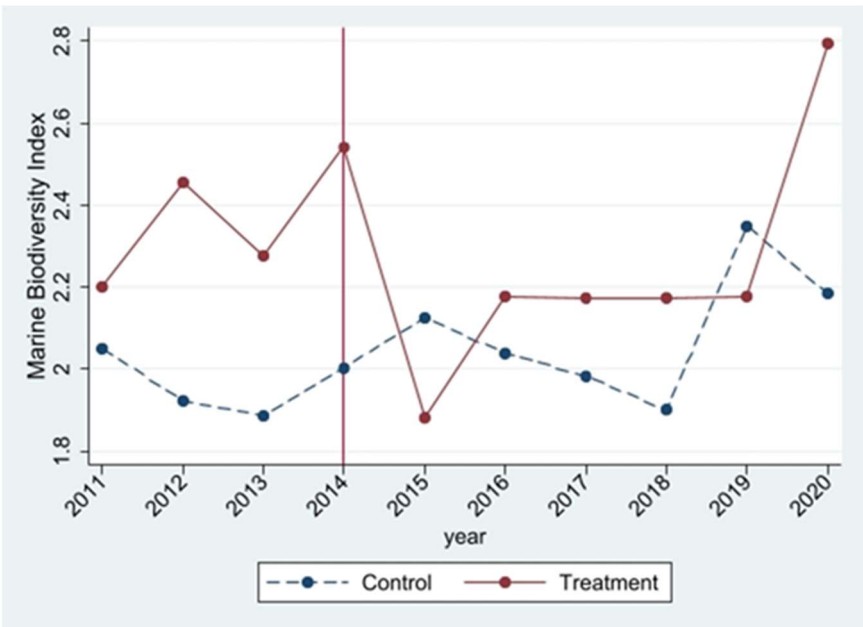

**Figure 2.** Parallel trends graph.

The abnormal trend after 2019 was caused by the Ministry of Ecology and Environment only counting the summer marine biodiversity index. In addition, we should distinguish between gold exploration and gold mining. Gold exploration is mainly on the sea surface, while gold mining is mainly through seabed mining, nearly 1 kilometer deep in the seabed [22]. The two ways of affecting marine biodiversity are different. The sea surface exploration of gold mines directly affects marine life, while the effect of seabed mining of gold mines on marine life is mainly indirectly through chemical pollutants, etc.

### 3.2. SVAR Analysis

SVAR with constraints is difficult to solve. We follow the idea of Cholesky decomposition, set matrix *A* as a lower triangular matrix where the main diagonal elements are all 1, and set matrix *B* as a diagonal matrix, which is called the "Cholesky constraint" [13,17,18]. The constraints are written as:

$$A = \begin{pmatrix} 1 & 0 & 0 \\ \cdot & 1 & 0 \\ \cdot & \cdot & 1 \end{pmatrix}, B = \begin{pmatrix} \cdot & 0 & 0 \\ 0 & \cdot & 0 \\ 0 & 0 & \cdot \end{pmatrix} \tag{10}$$

The missing value "." represents a free parameter (that is, no constraint) [13,23,24]. In the above formula, it can be seen from the first row of matrix *A* that $y_{2t}$ and $y_{3t}$ have no direct influence on $y_{1t}$. Similarly, from the second row of matrix *A*, it can be seen that $y_{1t}$ has a direct effect on $y_{2t}$, but $y_{3t}$ has no direct effect on $y_{2t}$. Finally, from the third row of matrix *A*, it can be seen that $y_{1t}$ and $y_{2t}$ have a direct effect on $y_{3t}$. Using the setting of matrix *A*, we can set that gold mining impacts precious metal smelting output value and marine product output. Precious metal smelting output value affects marine product output. Precious metal smelting output value and marine product output have no impact on gold mining. Relevant data are processed by logarithmic first-order difference processing. The constraints imposed on matrices *A* and *B* are called "short-run restrictions" and the SVAR model is called "short-run SVAR" [25]. We take the first-order lag for short-run SVAR. The results are shown in Table 2. All parameters are accurately identified.

To be more specific, all the SVAR coefficients here need to be interpreted in shifting terms, which means positive coefficient means negative effect and vice versa. The order of the three indicators is "gold mining", "precious metal smelting output value", and "marine product output". Thus, $A_{21}$ (2_1 in Table 2) of matrix *A* means gold mining signif-

icantly and negatively impacts precious metal smelting in a one year lag. This could be understandable in that resource-based monopolies will protect and adjust their production capacity to maintain resource savings. If they exploit more in the previous year, they may control production capacity in the next year. $A_{31}$ (3_1) means gold mining negatively impacts marine products but not significantly. $A_{32}$ (3_2) means pollution generated during the gold mining negatively impacts marine products significantly. In short, the change in gold production significantly and negatively impacts precious metal smelting but not for marine products.

**Table 2.** Identified results of short-run SVAR model.

| Sample: | 2013 Thru 2019 | Std. Err. | z | Number of Obs = | 7 |
|---|---|---|---|---|---|
| Exactly | Identified Model Coefficient | | | Log Likelihood = P > z (95% Conf. | 41.46849 Interval) |
| /A | | | | | |
| | 1_1 1 | (constrained) | | | |
| | 2_1 4.850272 | 1.880755 | 2.58 | 0.010 1.164059 | 8.536485 |
| | 3_1 0.2231218 | 0.244926 | 0.91 | 0.362 −0.2569241 | 0.703168 |
| | 1_2 0 | (constrained) | | | |
| | 2_2 1 | (constrained) | | | |
| | 3_2 0.113343 | 0.035247 | 3.22 | 0.001 0.0442598 | 0.182426 |
| | 1_3 0 | (constrained) | | | |
| | 2_3 0 | (constrained) | | | |
| | 3_3 1 | (constrained) | | | |
| /B | | | | | |
| | 1_1 0.0254109 | 0.006791 | 3.74 | 0.000 0.0121001 | 0.038722 |
| | 2_1 0 | (constrained) | | | |
| | 3_1 0 | (constrained) | | | |
| | 1_2 0 | (constrained) | | | |
| | 2_2 0.126445 | 0.033794 | 3.74 | 0.000 0.0602103 | 0.19268 |
| | 3_2 0 | (constrained) | | | |
| | 1_3 0 | (constrained) | | | |
| | 2_3 0 | (constrained) | | | |
| | 3_3 0.0117917 | 0.003152 | 3.74 | 0.000 0.0056149 | 0.017968 |

Another type of SVAR constraint is "long-run restrictions"; that is, to restrict the long-term effect of structural shock $\varepsilon_t$ on $y_t$, which was initiated by Blanchard and Quah [19]. This type of SVAR model is called long-run SVAR [26]. The constraints can be written as:

$$C = \begin{pmatrix} \cdot & 0 \\ \cdot & \cdot \end{pmatrix} \tag{11}$$

Here we only care about the long-term impact of gold production on the production of marine products. This constraint indicates that the production of marine products has no effect on the production of gold, and other parameters can be freely estimated. We also choose the first-order lag. The specific results are shown in Table 3. In the long run, the impact of gold production on marine products is still significantly negative. All the coefficients are significant. $A_{21}$ *and* $A_{22}$ *of* matrixC mean the influences of gold production and marine product output itself on marine products. In the long run, gold production and marine product output itself will adversely affect marine products. The short-long analysis shows a potential production-pollution-damage causality to marine life in time series.

**Table 3.** Identified results of long-run SVAR model.

| Sample: | 2013 Thru 2019 | Std. Err. | z | Number of Obs = | 7 | |
|---|---|---|---|---|---|---|
| Exactly | Identified Model Coefficient | | | Log Likelihood =<br>P > z | 29.55464<br>(95% Conf. | Interval) |
| /C | | | | | | |
| | 1_1 0.0648288 | 0.0173262 | 3.74 | 0 | 0.03087 | 0.098788 |
| | 2_1 0.0292612 | 0.0111069 | 2.63 | 0.008 | 0.007492 | 0.05103 |
| | 1_2 0 | (constrained) | | | | |
| | 2_2 0.020867 | 0.0055769 | 3.74 | 0 | 0.009936 | 0.031798 |

## 4. Discussion

Sustainable development maximizes resource conservation and rational use, minimizes the adverse effects of human production and consumption behavior on the environment, and achieves the coordination and balance of economic growth, natural resources, and the ecological environment [27–29]. As marine resources emerge with abundant reserves, the ocean's development has become an essential part of countries' sustainable economic growth [30,31]. However, we cannot sacrifice marine ecology for short-term benefits. Properly balancing the exploitation of marine mineral resources and the protection of marine biological resources is of great significance to the sustainable development of China's maritime economy.

### 4.1. Reducing the Impact of Marine Mineral Resource Exploration

The mining of marine mineral resources might influence tiny marine organisms' standard life patterns, such as spawning, growth, migration, etc. and further cause environmental pollution, including releasing COD and nitrogen oxides into the sea, resulting in the death of large marine organisms, while many marine exploration activities might be radioactive (for instance, the flipping of marine sediments may lead to the release of certain radioactive elements), impacting aquatic life and destroying the original ecology [32–40]. Therefore, in marine development, the relationship between the exploration of marine mineral resources and the protection of the marine ecological environment should be balanced. The local standards for pollutants discharged into the ocean should be stricter [41,42]. The direct toxic effects of heavy metals, petroleum, pesticides, solid waste, and other contaminants and the eutrophication caused by organic substances and nutrients such as COD and nitrogen oxides should be considered.

In particular, it should be pointed out that the impact of marine mineral exploration activities on marine diversity still cannot be ignored. According to Section 3.1, the whole biodiversity of Laizhou Bay decreased by 0.27 percentage even purely due to marine mining in Sanshan Island-Jiaojia.

### 4.2. Improve the Maintenance Level of Marine Biological Resources

Unlike the various minerals, rocks, and fossil fuels in marine resources, which are very slow to regenerate or almost impossible to revive, marine biological resources are limitedly renewable. According to the results of Section 3.2, in the long run, fishing for marine products will have a 2.1% adverse effect on marine products themselves. If marine species gradually shrink and disappear, there will be disastrous consequences. Nevertheless, as the maintenance level improves, its value will continue to increase [43–46]. Therefore, scientific and reasonable utilization and protection of species provenance and scientific management are required.

### 4.3. Paying Attention to the Long-Term Effects of Pollution and Strengthening Scientific Research

The diversity indicators used in marine yearbook statistics are mainly phytoplankton, general plankton, benthic organisms, and other relatively low-level and sensitive organisms. Therefore, exploration and small-scale trial mining at that year or earlier can have a very

significant impact. In the same way, the eutrophication of seawater has a direct and rapid effect on such organisms. As for the biological niches such as large fishes, the impact of marine mineral mining on them is reflected in an enrichment process in the marine ecosystem. At the same time, the transmission of pollutants caused by marine mineral mining also has an inevitable diffusion process, so the ecological effect may not be reflected immediately in the short-term [47–51]. According to the results of Section 3.2, in the short-term, gold production prima facie seems not to be related to the marine environment. In the long run, gold production will have an apparent 2.9% adverse effect on marine products. Therefore, long-term scientific research and tracking should be strengthened. In this regard, relevant research on the sustainable utilization of marine mineral resources is still scarce in China or even in the world. This article has laid a preliminary scientific foundation for quantifying the development of marine mineral resources and hopes to attract more global attention to the impact of land-based human activities on the marine environment and resources.

## 5. Conclusions

The blue economy accounts for 5% of global GDP. Coastal resources such as fish, minerals, and energy are vital to a sustainable blue economy, but they are affected by land-based human activities on a large scale. Decision-makers must transform the existing land–sea governance structure, strengthen the collaborative governance between land-based human activities and marine resources, and seek a new method of governance [52]. In marine development, the relationship between the exploration and development of marine mineral resources and the marine ecological environment should be dealt with first, and then we should strive to form an excellent navigational resource development system. We hope that this research can contribute to the UN Ocean Conference held in Lisbon, Portugal, in 2022.

**Author Contributions:** Conceptualization, S.Z. and Y.Y.; methodology, Y.Y., S.Z., G.H. and Z.W.; formal analysis, Y.Y. and R.Y.; writing—original draft preparation, Y.Y. and S.Z.; writing—review and editing, S.Z.; supervision, S.Z.,Y.Y., M.X., G.H., R.Y. and Z.W. collected and analyzed data, contributing equally as the second authors. All authors have read and agreed to the published version of the manuscript.

**Funding:** This research received Overall planning of land and space in Yangquan City (2020–2035) Grant YQZC20201337.

**Institutional Review Board Statement:** Not applicable.

**Informed Consent Statement:** Not applicable.

**Data Availability Statement:** All data are included in the paper or could be openly obtained.

**Acknowledgments:** The authors would like to thank the constructive comments and engagement with the paper from our reviewers and colleges in relation to significance to the world and choice of variables.

**Conflicts of Interest:** The authors declare no conflict of interest.

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
