# Peer review of "The Sustainable Development Path of the Gold Exploration and Mining of the Sanshan Island-Jiaojia Belt in Laizhou Bay: A DID-SVAR Approach"

_sustainability, doi:10.3390/su132111648_

Round 1

Reviewer 1 Report

General comments:

The Authors made important changes to the manuscript and adapted the title to the subject of the study. Overall, this resulted in a significant increase in the credibility and clarity of the manuscript.

However, they did not avoid mistakes that are not trivial.

Detailed comments:

Figure 1. Add a north sign and a linear scale

Page 2 line 80 , instead quaternary, change Quaternary deposits or sediments

Page 2 line 81-83, in geology, the oldest rocks are first described, deep, and then towards the surface. Swap the order of sentences.

Page 9 line 346-349, in what period did biodiversity decline? Whether it lasted a year or more, the graph (Figure 2) does not mean that it needs to be clarified.

Page 10, line 372, explain what research is so important to the world: DID SVAR, Sustainable development of gold exploration maybe impact of marine environment, the sentence provokes a check 

Author Response

General comments:

The Authors made important changes to the manuscript and adapted the title to the subject of the study. Overall, this resulted in a significant increase in the credibility and clarity of the manuscript.

However, they did not avoid mistakes that are not trivial.

Response: We greatly appreciate your constructive comments, and we have prepared a detailed reply to your comments.

Detailed comments:

Figure 1. Add a north sign and a linear scale

Response: Thanks a lot for your comment. According to your suggestion, we have remade the corresponding map. (line 75 in the latest updated manuscript) 

Page 2 line 80 , instead quaternary, change Quaternary deposits or sediments

Response: We are grateful to Reviewer 1 for pointing out this issue. We have adjusted the text in line 79 in the latest updated manuscript.

Page 2 line 81-83, in geology, the oldest rocks are first described, deep, and then towards the surface. Swap the order of sentences.

Response: We greatly appreciate the reviewer's comments. We have rearranged the text in lines 78-81.

Page 9 line 346-349, in what period did biodiversity decline? Whether it lasted a year or more, the graph (Figure 2) does not mean that it needs to be clarified.

Response: We sincerely agree with the reviewer. In the revised manuscript, one sentence has been added to further clarify this question:

"Although partly recovered in 2016 and subsequent years, the marine biodiversity index has not yet returned to its pre-2015 level, except for the outliers caused by changes in the statistical caliber in 2019." (Lines: 267-269)

Page 10, line 372, explain what research is so important to the world: DID SVAR, Sustainable development of gold exploration maybe impact of marine environment, the sentence provokes a check 

Response: We are very sorry that we failed to explain our views clearly. We have revised the contents of this part in lines 375-376. Also, one sentence has been added: 

“This article has laid a preliminary scientific foundation for quantifying the development of marine mineral resources and hopes to attract more global attention to the impact of land-based human activities on the marine environment and resources.” (Lines 377-380)

Once again, thank you very much for your suggestion. We really appreciate your help in improving the quality of this manuscript. We would be glad to reply to any further questions and comments that you may have. 

Reviewer 2 Report

This manuscript tries to employ DID/SVAR models to understand the impact of gold mining industry's activities onto surrounding marine ecosystem. Topic sounds interesting and methodologies seems OK. Compared to the previous version, now the results from the DID/SVAR models are clearly stated and utilized in the discussion. Overall, the manuscript seems OK except for the following small points for the Figure 2. 

In Figure 2, the authors put the title (?) of the figure above the graphic (Parallel Trend Chart) though the true caption exists below the figure, of course. The authors should somehow delete this unneeded title.

Author Response

Response: We are incredibly grateful to Reviewer 2 for pointing out this problem.  The redundant title in Figure 2 has been deleted. Thank you very much for your suggestions. 

Reviewer 3 Report

The manuscript has been improved according to the comments and can be accepted for publication.

Author Response

Response: Thank you for your kind consideration of our manuscript. We will continue to improve the quality of the manuscript. We would be glad to respond to any further questions and comments that you may have.

This manuscript is a resubmission of an earlier submission. The following is a list of the peer review reports and author responses from that submission.

Round 1

Reviewer 1 Report

The discussed issue is extremely important and essential. The impact of mining on an environment as sensitive as the marine one is worthy of any study and approximation. Unfortunately, after reading the article, I have to state significant discrepancies between the title, the subject matter and the content of the article.

The first word of the title is Ecology. This is the science of the ecosystem. Even the generally complex mores systems around Sheshen Island are not explained in this article. Based on their explanations, it is not even possible to find the island of Shenshen in the Laizhou Bay. The lack of a figure with the location of the research is a signal that we are dealing with very theoretical work and the model used is often found in financial and statistical applications.

The methodology chapter covers the stages of the model being prepared in detail, but there is no information about the data that will be used to assess the ecological impact. What environmental indicators. What does it mean, data on marine biodiversity come from the statistical yearbook (page 3 line 90-93)???? In a word, the chapter does not allow the reader to verify the final result obtained. Which means marine biodiversity is revealed only in the chapter discussion does not arouse my confidence that these are facts.

Figure 1 appears as the result (not referred to anywhere in the text), which contains the explanation "total" for Y and I do not know what? Is this the value of marine biodiversity? Interestingly, after 2014 (opened of gold mining) and a clear decrease, this value returns to the initial level in the following years and even grows even higher (total). If so, then maybe the gold mine increases biodiversity after 4 years of operation?

The discussion contains fairly general statements that cannot be argued with. I find no connection here to the results the authors received

In my opinion, the article does not correspond to the topic covered in the title. It contains numerous mental shortcuts, and the results cannot be verified. It is not printable at its current level.

Reviewer 2 Report

This manuscript tries to apply DID/SVAR model to understand the impact of gold mining industry's activities onto surrounding marine ecosystem. The topic/objective/methods are somewhat acceptable. However, one concern is the numerical result is not much discussed, I guess. Especially for the result from SVAR model, the results are shown in Table 2 and 3, but almost no discussion on the values. I do understand that discussing all the estimated parameters is difficult and maybe meaningless, but still the authors should utilize the numerical results they obtained.
Due to the shortcomings in the result section, discussion and conclusion section is quite qualitative with almost no support by quantitative results. 
Therefore, I do recommend the authors to revise the manuscript by including more quantitative contents in Results section to have more scientifically defendable discussion/conclusion section.

Followings are the minor points.

Abstract: You should spell out DID-SVAR here, or where it appears for the first time in the main text.
Somewhere in the method section: The authors should add the information of the programs or packages they used. 
Fig 1.: Simply, the quality of the figure should be bettter than this.
L. 214-221: Is there any reason why this paragraph is in italic fonts?
Table 2: The meaning of "1_1", "2_1", ... is a bit difficult to understand. Considering the broad audience of this journal, you'd better add some explanation.
Table 3: Is this result for the short-run or long-run. If former, where is the result for long-run?

Reviewer 3 Report

Authors performed research focused on application of DID-SVAR Approach on mine ecological impacts.

Good introduction supported by adequate and recent references. Aims clearly stated.

On line 78, DID-SVAR is first time presented on text so it must be (shortly) explained (presently, only along 2.3 and 2.4).

Methodological approach is adequate and correct, with proper supporting references.

Results presentation is, in general, clear and sound, supported by figures.

Discussion can be improved comparing obtained results with similar ones on adequate references.

Conclusions are coherent with obtained results but can be strengthened after improving discussion as suggested.